# Silicon as a ubiquitous contaminant in graphene derivatives with significant impact on device performance

Rouhollah Jalili [1], Dorna Esrafilzadeh[2], Seyed Hamed Aboutalebi [3,4,5], Ylias M. Sabri[6], Ahmad E. Kandjani[6], Suresh K. Bhargava[6], Enrico Della Gaspera [1], Thomas R. Gengenbach[7], Ashley Walker [8], Yunfeng Chao[8], Caiyun Wang[8], Hossein Alimadadi[9,10], David R.G. Mitchell[11], David L. Officer [8], Douglas R. MacFarlane[12] & Gordon G. Wallace[8]

Silicon-based impurities are ubiquitous in natural graphite. However, their role as a contaminant in exfoliated graphene and their influence on devices have been overlooked. Herein atomic resolution microscopy is used to highlight the existence of silicon-based contamination on various solution-processed graphene. We found these impurities are extremely persistent and thus utilising high purity graphite as a precursor is the only route to produce silicon-free graphene. These impurities are found to hamper the effective utilisation of graphene in whereby surface area is of paramount importance. When non-contaminated graphene is used to fabricate supercapacitor microelectrodes, a capacitance value closest to the predicted theoretical capacitance for graphene is obtained. We also demonstrate a versatile humidity sensor made from pure graphene oxide which achieves the highest sensitivity and the lowest limit of detection ever reported. Our findings constitute a vital milestone to achieve commercially viable and high performance graphene-based devices.

[1] School of Science, RMIT University, Melbourne, VIC 3001, Australia. [2] School of Engineering, RMIT University, Melbourne, VIC 3001, Australia. [3] Institute for Superconducting and Electronic Materials, Australian Institute for Innovative Materials, University of Wollongong, Wollongong, NSW 2522, Australia. [4] Condensed Matter National Laboratory, Institute for Research in Fundamental Sciences, Tehran 19395-5531, Iran. [5] Pasargad Institute for Advanced Innovative Solutions (PIAIS), 1991633361 Tehran, Iran. [6] Centre for Advanced Materials and Industrial Chemistry (CAMIC), School of Science, RMIT University, Melbourne, VIC 3001, Australia. [7] Manufacturing, Commonwealth Scientific and Industrial Research Organisation, Clayton, VIC 3168, Australia. [8] Intelligent Polymer Research Institute & ARC Centre of Excellence for Electromaterials Science, Australian Institute for Innovative Materials, University of Wollongong, Wollongong, NSW 2522, Australia. [9] DTU Danchip/Cen, Technical University of Denmark, Center for Electron Nanoscopy, Fysikvej, Building 307, 2800 Kgs. Lyngby, Denmark. [10] Danish Technological Institute, Kongsvang Alle 29, 8000 Aarhus C, Denmark. [11] Electron Microscopy Centre, Australian Institute for Innovative Materials, University of Wollongong, Wollongong, NSW 2522, Australia. [12] ARC Centre of Excellence for Electromaterials Science, Monash University, Clayton, VIC 3800, Australia. Correspondence and requests for materials should be addressed to R.J. (email: ali.jalili@rmit.edu.au) or to D.E. (email: Dorna.esrafilzadeh@rmit.edu.au)

arge-scale, cost-effective methods for producing high-quality and functional atomically thin two-dimensional (2D) materials with unmatched properties are critical to realising commercial applications of these promising materials[1–5]. However, the current top–down synthesis methods are prone to contamination, which can adversely affect properties, such as optical absorption and emission, electrochemical properties, carrier mobility, biological activity and toxicological properties[6–13]. These contaminants can cause inferior or unpredictable properties and are at the heart of many seemingly inconsistent reports on the properties of 2D materials[8,14,15]. For instance, several contradictory reports on electrochemical properties, electrocatalytic activity and biological properties of graphene and graphene oxide (GO) were ultimately understood to be the result of metallic impurities or acidic residues[7,8,16–20]. Such inconsistency prevents the development of a robust regulatory framework governing the implementation of such layered nanomaterials, especially if they are destined to become the backbone of next-generation devices. Perhaps, more importantly, this has stymied the emergence of a major application or the so-called "killer app" for graphene-based systems[21], a long-promised but as-yet unrealised goal. Graphene may therefore follow the same trajectory of carbon nanotubes[14,21], which were once billed as being transformative but have so far failed to make a significant commercial impact.

The need for methods that can produce higher quality and purity graphene and other 2D-based materials is widely recognised for device fabrication. Despite this, there seems to have been little effort devoted to understanding what effect impurities are having on the performance. The graphene glut in recent years has resulted in an increase in the use of cheaper alternative feedstocks[21]. However, to date, no rigorous quality-control mechanism has been put in place to understand the detrimental effects of impurities on the final performance of devices, mainly due to the lack of knowledge of the types of impurities and their potential influence. As graphene is a 2D material system, which is basically only a surface, any contaminant can alter its intrinsic properties dramatically by affecting surface active sites[6,14] and/or by reducing the available surface area, as shown schematically in Fig. 1. This can result in decreased performance in active surface area-dependent applications. Moreover, this also indicates the possibility that the as-exfoliated graphene layers can each have different properties, again highlighting the importance of pure 2D materials. For instance, the adulteration of carbon bonds in real-world, two-electrode supercapacitors severely affects the double-layer capacitance[22]. This leads to capacitance values much lower than the theoretical capacitance of graphene (550 F/g)[23], with

values of 409 F/g being the highest reported to date[24]. As the production capacity of graphene and other 2D materials grows exponentially, the demand for high-performance 2D material-based devices will become significant[2,21]. This calls for a broad effort to identify the types and extent of impurities present and the best way to deal with them to achieve the maximum potential of the broad family of 2D materials.

Here direct visualisation of the surface of various solution-processed graphene reveals the existence of significant amount of silicon-based contaminations at the atomic level. High-angle annular dark field (HAADF) imaging combined with energy-dispersive X-ray spectroscopy (EDS) in an aberration-corrected scanning transmission electron microscope (STEM) was employed to detect and image these impurities. The present study addresses the challenge of obtaining high-purity graphene and GO from graphite (top–down approach) and offers a broad and comprehensive perspective on how to dramatically enhance the final performance of such materials in diverse applications. We identify the most prominent factor impacting the performance of graphene-based devices, namely inherent contaminants arising from impure graphite feedstock. Ultra-high performance humidity sensors and supercapacitors from graphene materials were then achieved through eliminating the adverse impact of endogenous silicon impurities.

## Result

**Atomic resolution observation of silicon impurity on graphene.** The presence of molecularly dispersed silicon-based contamination was not evident through high-resolution TEM bright-field (BF) analysis of GO samples (Fig. 2). Monodisperse, amorphous materials are typically very difficult to resolve in conventional phase-contrast high-resolution TEM/STEM imaging particularly in the presence of contaminants[25]. However, the situation changes dramatically when using HAADF imaging. The atomic resolution capabilities of aberration-corrected STEM permits single-atom imaging in HAADF mode by virtue of its high atomic number sensitivity (the contrast is roughly proportional to $Z^2$, where $Z$ is the mean atomic number)[26]. Therefore, the higher atomic number of silicon with respect to carbon and oxygen ensures that Si (and other high atomic number element) atoms are visible as bright spots in HAADF micrographs, while this was not possible with conventional BF images. Figure 2 shows silicon-based contamination as molecules and clusters thereof cover a large fraction of the surface area of GO, which produced from graphite of low purity (i.e. 98%). The existence of these impurities is also further verified by performing EDS in

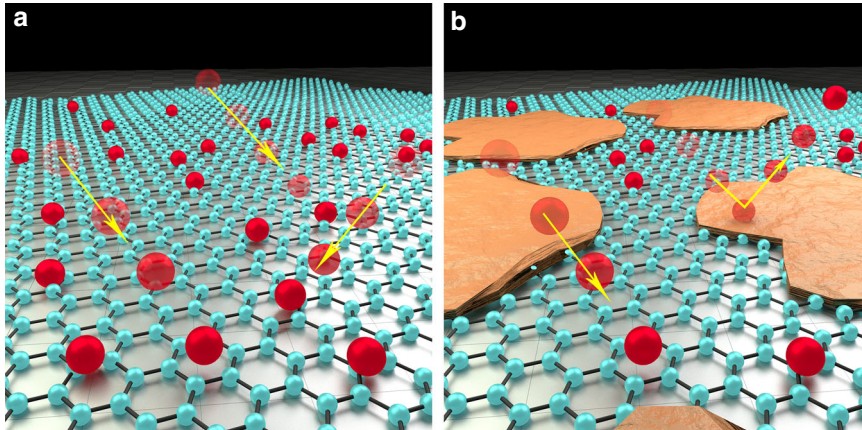

**Fig. 1** Schematic representation of the available surface area of graphene for molecular interaction. **a** Pure surface vs **b** contaminated surface. The red spheres represent molecules that can interact with the surface, while the orange rafts represent the contaminants

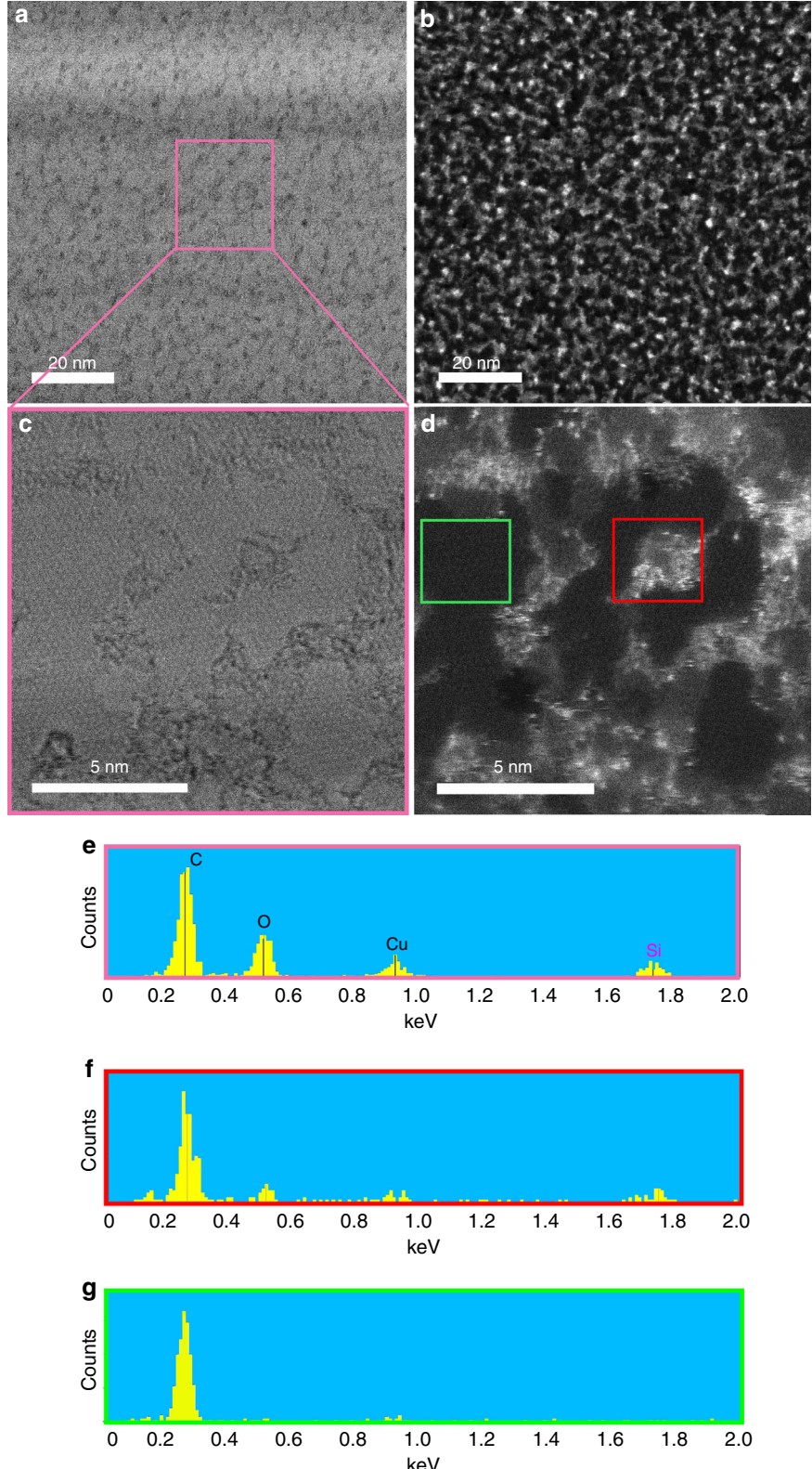

**Fig. 2** The extent of silicon-based contamination on the surface of typical graphene oxide derived from low-purity graphite (98% purity). **a** Bright-field (BF) image of a typical GO sheet. **b** HAADF image of **a**. **c**, **d** Details of BF and HAADF images of the marked region in **a** at higher magnification, respectively. Unlike the BF images in which Si contaminants are largely invisible, the HAADF images highlights them as bright clusters. **e** EDS spectrum of the entire region shown as pink box in **a**, **c**. The strong Si peak at 1.739 keV confirms the significant contamination in the GO sample. **f**, **g** A comparison of the EDS spectra of the contaminated area (**f**) and non-contaminated area (**g**), which are marked as red and green boxes in **d**, respectively

parallel with HAADF on the same region. The EDS spectrum of GO sheet (Fig. 2e) identifies a significant amount of silicon-based contamination. The peaks at ~0.277, 0.525 and 1.739 keV in the EDS spectrum are due to C, O and Si, respectively, while the peak at 0.930 keV is from the Cu (support) grid. Comparing the EDS spectra of two neighbouring regions, one clean (dark) and the other bright (contaminated) confirms silicon to be the contaminant (Fig. 2 f, g and Supplementary Figure 1). The contaminated region (red boxed region in Fig. 2d) showed a noticeable peak at 1.739 keV (Fig. 2f), while the clean regions (green box in Fig. 2d) showed no such silicon peak (Fig. 2g).

Oxidative exfoliation of graphite, i.e. modified Hummers' method, was used here[27,28] and requires several chemical treatment steps any, or all of which, could contribute to the observed silicon-based contamination. However, the impurity was also present in solvent-exfoliated graphene layers prepared by bath sonication of graphite powder in a very pure exfoliating solvent (Fig. 3 and Supplementary Figure 2). Solvent exfoliation of graphite uses a solvent (ca. N,N-dimethylformamide) for the

exfoliation process to give graphene in the liquid phase (monolayer and few layers) without any additional oxidation step[29]. This showed that silicon-based compounds are ubiquitous contamination in graphene-based materials when using top–down production approaches and is not caused solely by reagents or particular chemical processes (i.e. modified Hummers' method used here[27,28]). Therefore, the silicon contamination originated from the graphite precursor.

HAADF imaging of the parent graphite (98% purity) demonstrated a significant amount of silicon-based contamination (Fig. 4). Detailed images of the three different subareas highlighted in this figure are shown in Fig. 3b–d. EDS shows regions to be iron-contaminated, clean and silicon-contaminated (Fig. 4e–g, respectively). Clean regions showed a perfect graphitic lattice structure with very little or no silicon presence, whereas other areas showed intractable and widespread silicon-based contamination along with some iron clusters. Natural graphite is mined and then purified using floatation. The purification in this process is based on

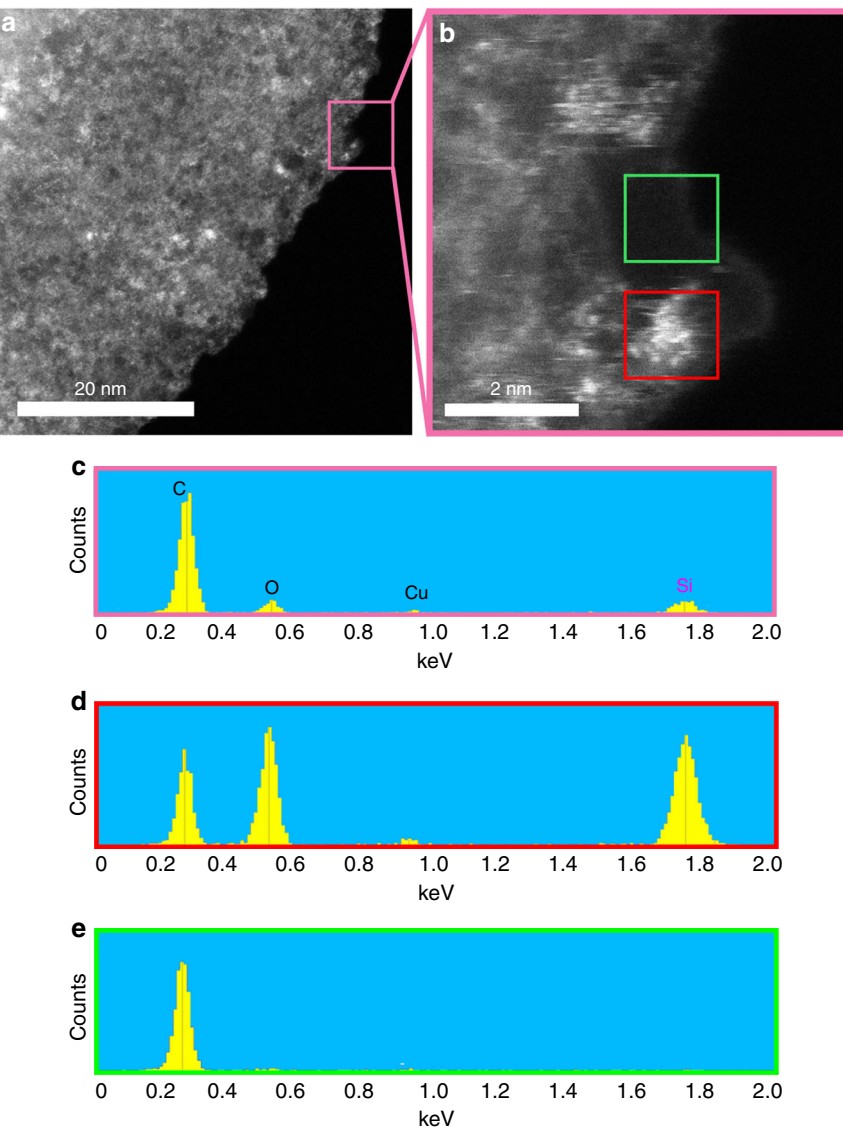

**Fig. 3** The extent of silicon contamination on the surface of typical solvent-exfoliated graphene derived from low-purity graphite (98% purity). **a** HAADF image of a typical graphene sheet. **b** Detail of HAADF image of the boxed region in **a**. **c** EDS spectrum of the boxed region in **a**. The strong Si peak at 1.739 keV confirms the presence of significant contamination. **d**, **e** A comparison of the EDS spectra of the contaminated area (**d**) and non-contaminated and monolayer area (**e**), which are marked as red and green boxes in **b**, respectively

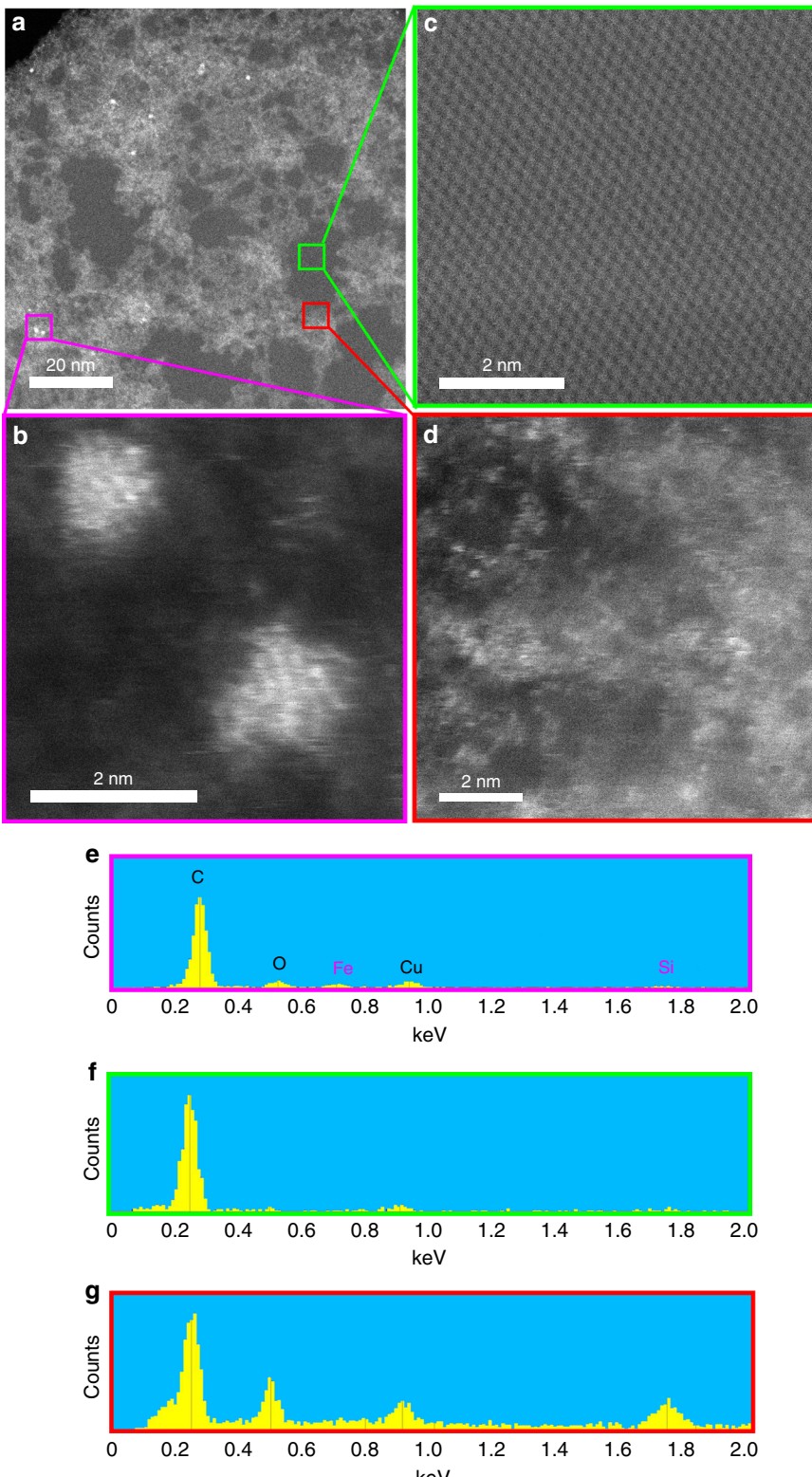

**Fig. 4** The extent of silicon contamination on the surface of typical low purity graphite (98% purity). **a** HAADF image of a typical graphite platelet. Details of the various boxed regions in **a** showing: **b** an iron contamination, **c** a clean area with a perfect graphitic lattice structure, and **d** a silicon contaminated area. **e**–**g** EDS spectra of **b**–**d**, respectively, showing iron contamination, clean graphene and silica contamination, respectively

differences between the surface chemistry of soil rock and graphite mineral[30]. However, the floatation process is not able to remove high abundance mineral impurities, such as silicon. These impurities are commonly removed by using chemical or thermal treatments[31]. Graphite particles in the purity range of 80–98% are typically refined using only floatation. For purities >98%, additional refinement steps are carried out following floatation[32]. This provides two options to eliminate the contamination: (a) purification of the exfoliated materials and (b) employing purer graphite precursors.

**Producing high-purity graphene**. As important as the removal of this contamination on the surface of GO is, it proved to be an almost impossible task (Fig. 5). Various methods were evaluated including extensive washing of the as-prepared GO material with boiling 5 M NaOH (Fig. 5a, b). However, the silicon-based contaminants proved to be persistent and appeared to become more widely dispersed across the surface. Purification with such a strong basic solution resulted in an irreversible agglomeration and restacking of GO sheets (Supplementary Figure 3). Consequently, the impurities are confined between the layers and remain following the purification process. Even chemical–reduction of GO proved to be unsuccessful in removing the impurities effectively (Fig. 5c). This, however, was not surprising as silicon–oxygen-rich compounds (i.e. silica) are typically considered to be corrosion-resistant materials and the only reagent that can effectively etch them is fluoride. However, even using $NH_4F$ to remove the impurities proved to be unsuccessful (Fig. 5d and Supplementary Figure 3–4), and this also resulted in an irreversible agglomeration of GO layers. Generally, increasing the ionic strength or decreasing the pH of GO suspensions results in loss of the surface charge and restacking of GO particles then occurs[33]. Moreover, the set-up and the process parameters that need to be optimised for the removal of silicon impurities are complex and hazardous and result in a significant increase in the cost of production[34].

A better approach is therefore to improve the quality and purity of the feedstock and to avoid the use of inexpensive and contaminated feedstocks, which are now typically used in non-research applications. Evaluation of various GO produced from graphite with a range of purities (98% to 99.9999%) revealed that purities of ≥99.9% result in almost contaminant-free GO (Fig. 6a–e & Supplementary Figure 5). Interestingly, a commercially obtained GO material, which was tested as a control, showed very significant silicon-based contamination. Furthermore, EDS spectra of GO derived from graphite with a purity of ≥99.9% showed no detectable silicon-based contamination (Fig. 6g–k). Nevertheless, the HAADF images still showed very limited numbers of impurity atoms (bright dots) even in the very high purity GO (Fig. 6d, e & Supplementary Figure 6). It has been shown previously that oxidation of graphite introduces varying types of impurities into the graphene materials, and their origin can be traced to impurities within the chemical reagents used during the synthesis[7]. This was confirmed by analysing a typical solvent-exfoliated graphene, derived from high-purity graphite (Supplementary Figure 7) and solvent, which represented a very pure surface (Fig. 6f and Supplementary Figure 8). The HAADF imaging technique also revealed regions where multiple layers of GO were present as rafts or plateaus. Presence of such oxidised rafts has been suggested through indirect characterisation techniques before[15,17,18,35]. These oxidised rafts are brighter than the single sheet areas as the HAADF image contrast is a function of both $Z^2$ and thickness.

**Characterisation of the impurity**. X-ray photoelectron spectroscopic (XPS) measurements were performed to further characterise the silicon-based impurities. GO samples derived from graphite with two different purities, 98% and 99.9%, were prepared by drop casting on a gold-coated wafer. Since XPS is extremely surface-sensitive with a sampling depth of only a few nm, we decided to use XPS depth profiling in order to probe the chemical composition at different depths. The use of an Ar cluster source instead of a conventional Ar ion gun enables sputtering (etching) of a wide range of materials, including organic compounds, while minimising damage to the chemical structure during the sputtering process. A comparison of the high-resolution C 1s spectra of the two materials provided strong evidence that the GO in both cases was essentially identical (Fig. 7a, b). Similar levels of Si (0.09%) were detected on the surface of both GO samples with the binding energy of the Si 2p peak (102 eV) indicating the presence of an organosilicon compound rather than an inorganic Si oxide ($SiO_2$), which would be expected at 103.5 eV (Fig. 7c and Supplementary Figure 9)[36,37]. This compound was completely removed after only 30 s of etching (Fig. 7d), suggesting it to be a very thin layer of adsorbed surface contamination. In contrast to the low-purity sample, an extensive washing and careful handling of the high-purity GO resulted in the removal of this adsorbed surface contamination (Supplementary Figure 10–11). Subsequent etching revealed a clear difference between the two GO samples below the surface: in the case of the purer GO (99.9%), Si was never detected again above the detection limit of the technique (ca. 0.01 %), confirming the high purity of the material; in the case of the lower purity GO (98%), Si reappeared over the etching and was thereafter present at about 0.15 atomic%. The Si 2p peak position remained at about 102 eV, characteristic of Si-O and Si-C bonds. We also note that, even under the very mild etching conditions used, the bombardment of the GO surface with Ar clusters caused a significant reduction of the GO (Supplementary Figure 12–13), which is consistent with the literature[38]. Interestingly, the purer GO (99.9%) was reduced much more rapidly than the lower purity GO (98%), probably due to a more pristine and uncontaminated surface.

In order to evaluate the average amounts of silicon contamination in the bulk materials, wavelength dispersive X-ray fluorescence (WD-XRF) spectroscopy was used. Results similar to the XPS depth profiling measurement, were obtained with 0.04 ± 0.007 and 0.25 ± 0.01% silicon found in the pure and non-pure samples, respectively. Furthermore, silicon-based impurities adversely affected the photoluminescence (PL) property of the GO materials as shown in Fig. 7e, f and Supplementary Figure 14. The origin of the PL in GO is due to the electronic transitions among and between the non-oxidised carbon regions and the boundary of oxidised carbon atom regions[39]. It appears that silicon-rich impurities can effectively hinder this electronic transition as well as being a physical barrier on the GO functional groups. It should be noted that the size of graphite (Supplementary Figure 15) and the resultant GO sheets in both samples (contaminated and pure GO) were almost identical (Supplementary Figure 16) eliminating the association of the observed phenomena to any size effects. Other physical properties measured by ultraviolet–visible (UV-Vis), Raman, Fourier transform infrared (FTIR) spectroscopy and XRD spectra were almost identical among these GO samples (Supplementary Figure 17–20). This, together with the marked similarity in the C 1s spectra (see XPS discussion above) and the aforementioned equal size distribution, confirms that the chemical and physical properties between GO samples are indistinguishable, except for the presence of Si. As we will show in the following section, these silicon-based impurities play a pivotal role in affecting the performance of graphene-based devices.

## Discussion

Identifying atomically dispersed and relatively low molecular weight impurities in graphene and other 2D materials is very challenging, as they are not easily visualised by routine imaging methods. In the case of silicon-based impurities, in contrast to metallic impurities[11], indirect detection through electrochemical measurements is not feasible due to the lack of an electrochemical redox response. Other analytical methods, such as inductively coupled plasma mass spectrometry, are also not effective as

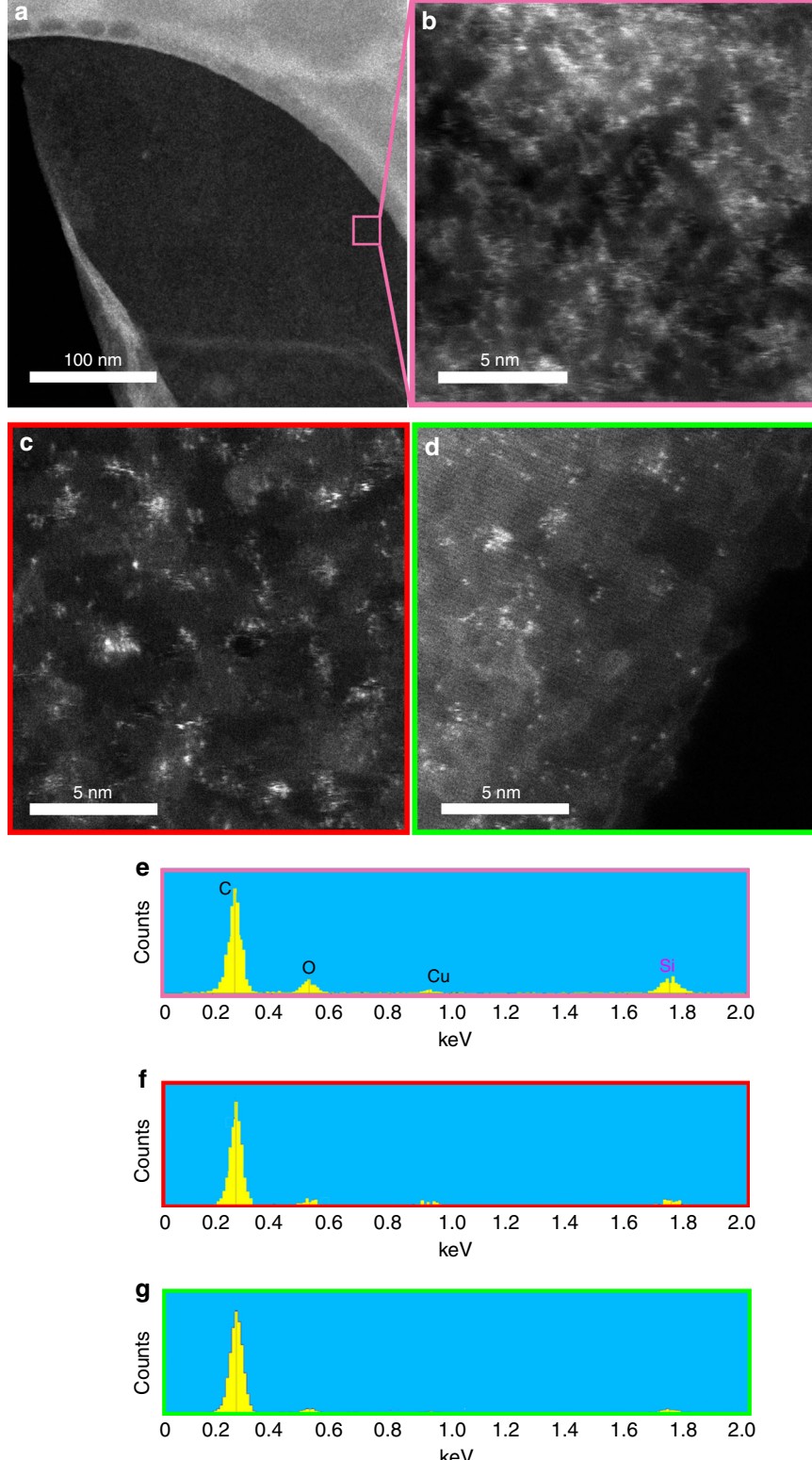

**Fig. 5** The effect of washing on typical graphene oxide derived from low-purity graphite (98% purity). **a, b** GO washed with 5 M NaOH at 120 °C. **a** Restacked GO sheets due to the basic washing. **b** Detail of the boxed region in **a** showing that the silicon-rich impurities have become more dispersed but have not been removed. **c** Chemically reduced GO showing the silicon-rich contamination. **d** NH$_4$F washed GO. The surface appears cleaner, but this treatment also causes significant agglomeration and restacking of sheets. **e–g** A comparison of the EDS spectra of the NaOH washed, chemically reduced and NH$_4$F washed GO in **b–d**, respectively

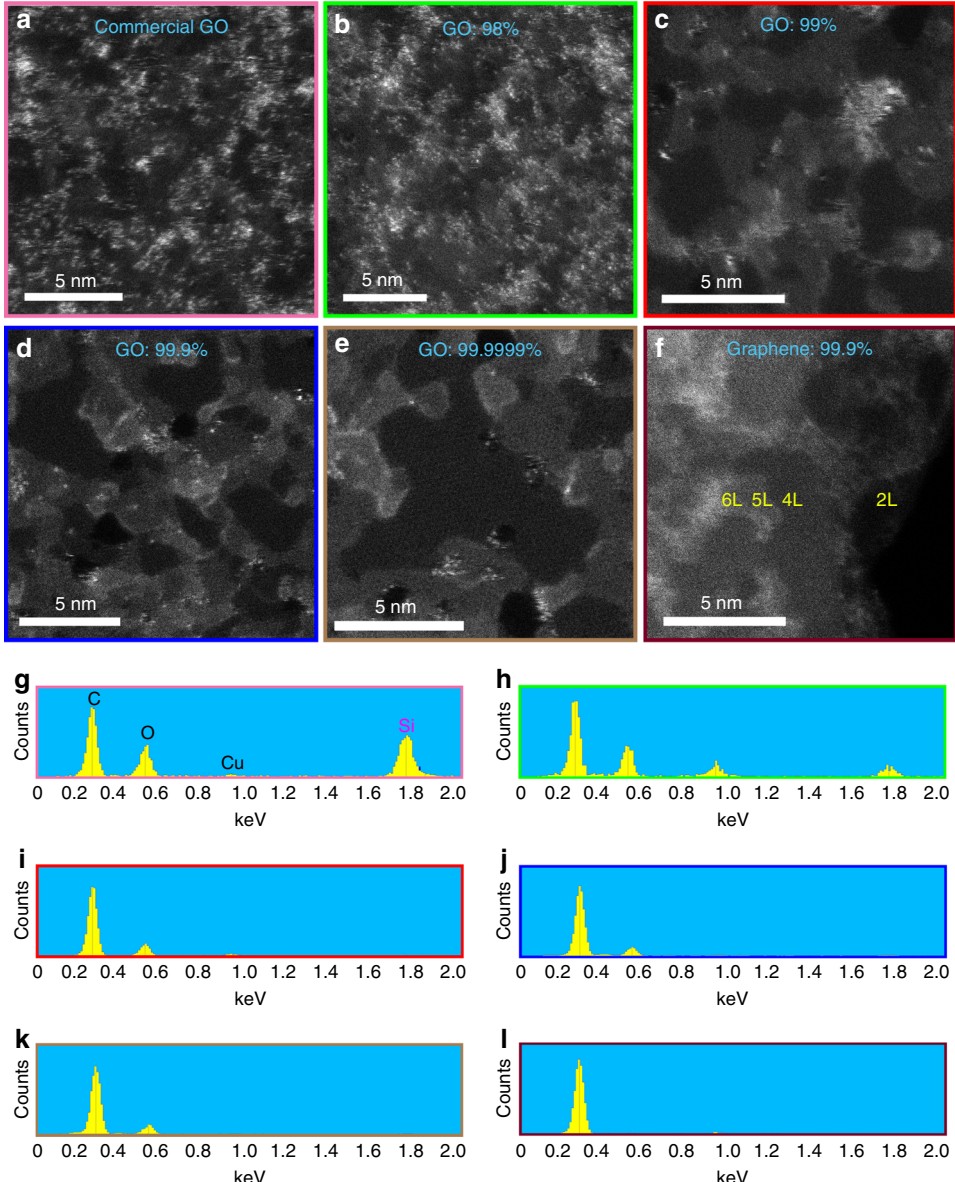

**Fig. 6** HAADF images of graphene and graphene oxide samples with varying degrees of purity. **a–e** Varying contamination degree on GO synthesised from various graphite feedstocks (of differing purity) along with a commercially purchased material: **a** commercially sourced GO; **b** natural graphite flake (98% purity); **c** natural graphite flake (99% purity); **d** natural graphite flake (99.9% purity); **e** natural graphite powder (99.9999% purity). **f** Typical solvent-exfoliated graphene synthesised from graphite with 99.9% purity showing a very clean surface with almost no contamination. Numbers of layers are marked on the image. Scale bars in the images are 5 nm. **g–l** Comparison of the EDS spectra of the samples shown at **a–f**, respectively. The panels are colour-coded for clarity

silicon-based compounds (i.e. silicon oxides) are unreactive in all acids except hydrofluoric acid (HF) and all three Si isotopes are subject to N- and O-based interferences[40]. These factors may explain why the existence of such ubiquitous contamination on solution-processed graphene, and other similar 2D materials, has not been previously reported in the literature.

It should be noted, although there are many methods to purify graphite such as hydrometallurgical purification methods (acid washing, floatation method) and pyrometallurgical methods (chlorination roasting method and the use of extremely high temperatures, >2700 °C), these methods are not applicable after the exfoliation in liquid media or device fabrication. In the case of hydrometallurgical purification methods, usually a purification level of up to 98% can be achieved. This method, although very efficient in the removal of metallic impurities, cannot be applied

to remove Si-based impurities. On the other hand, heating the graphene at high temperature (>2700 °C) is not a viable option in most cases, as this will limit the versatility of end-product device fabrication. These limitations are much more significant in the case of delicate dispersions of graphene and GO. As such, using high-purity graphite (commercially available) in the first place is recommended.

An important question is whether the impurities described here have any undesirable effect in the final performance of practical devices. Oxygen-containing functional groups on the hydrophilic surface of GO provide a high potential to adsorb water molecules[41]. Therefore, GO-based relative humidity (RH) sensors have shown great promise in this respect[42–44]. However, the detection range and moisture uptake reported so far, although promising, represent just an incremental improvement over

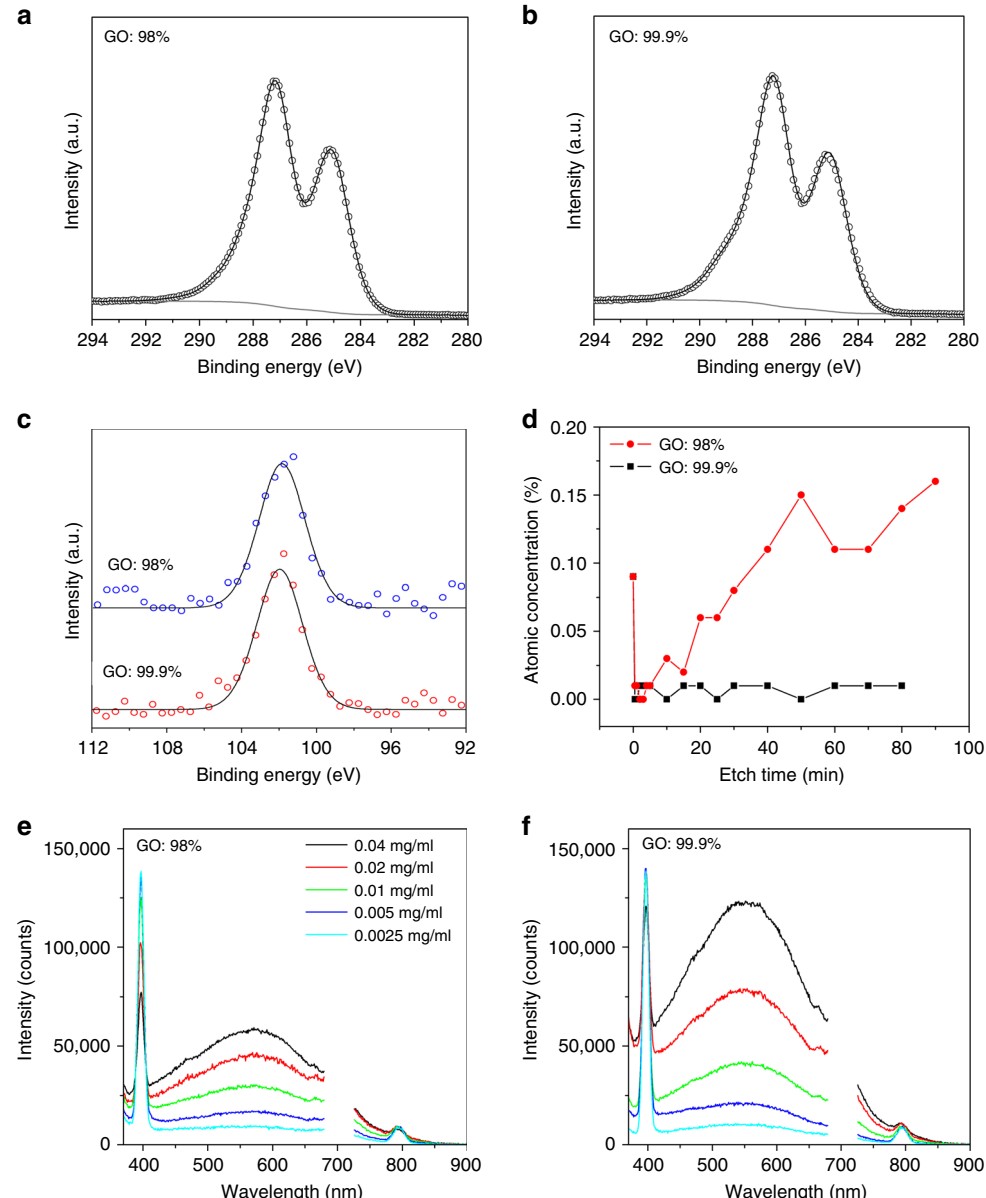

**Fig. 7** Characterisation of typical GO films and dispersions prepared from graphite feedstock of different purities. **a**, **b** Comparison of the XPS C 1s spectral region of GO films. **c** Comparison of the XPS Si 2p spectral region of GO films. **d** Comparison of the atomic concentration of silicon as a function of etching time. **e**, **f** Comparison of photoluminescence spectra ($\lambda_{exc} = 350$ nm) of GO dispersions in water as a function of solution concentration. The observed sharp peaks at 396 and 792 nm are due to the Raman peaks of water. The second-order diffraction peak at 700 nm has been removed for clarity

existing technologies. We anticipated that silicon-based contamination might act as a barrier, blocking the moisture absorber sites and thus adversely affect the sensor performance. Therefore, the first test of our hypothesis was to evaluate this effect by fabricating an RH sensor from a variety of GO materials.

Three almost identical thin films of GO derived from various feedstocks (a commercial GO and two derived in-house from graphite of 98% and 99.9% purities) were deposited on a quartz crystal microbalance (QCM) to fabricate the RH sensors. All QCM-based humidity sensors (Fig. 8a) revealed strong dynamic responses with excellent reproducibility towards even the lowest humidity level (see Supplementary Table 1). Notably, the GO prepared from the highest purity graphite (99.9%) showed a significantly higher sensitivity (66.5 Hz/% RH) when compared to that of less pure materials (53 Hz/% RH) (Fig. 8b). This sensitivity towards humidity was over two times higher than the highest

ever reported sensitivity of 28.7 Hz/% RH using a copper metal–organic framework as the sensitive layer (also see Table S1)[45]. Such high sensitivity results in an exceptionally high signal-to-noise ratio (~2000 Hz/Hz) thus allowing for trace levels of humidity to be detected. This high level of sensitivity corresponded to a very high humidity uptake, ranging from 7.5 wt% at 2.5% RH up to 32 wt% at 90% RH. The highest purity GO sheets displayed an extremely low limit of detection (LOD) of 0.006% RH at 27 °C, which is equivalent to 1.5 mg/m³ (2 ppm) absolute humidity; a level which has not been reported to date. This LOD is at least two orders of magnitude superior to the best-performing 2D MXenes (LOD of 0.8% RH) and around one order of magnitude better than the lowest reported LOD in the literature[46]. Similar trend was found when the effect of temperature is considered (Fig. 8c and Supplementary Figure 21). The GO sensor was also found to possess excellent repeatability

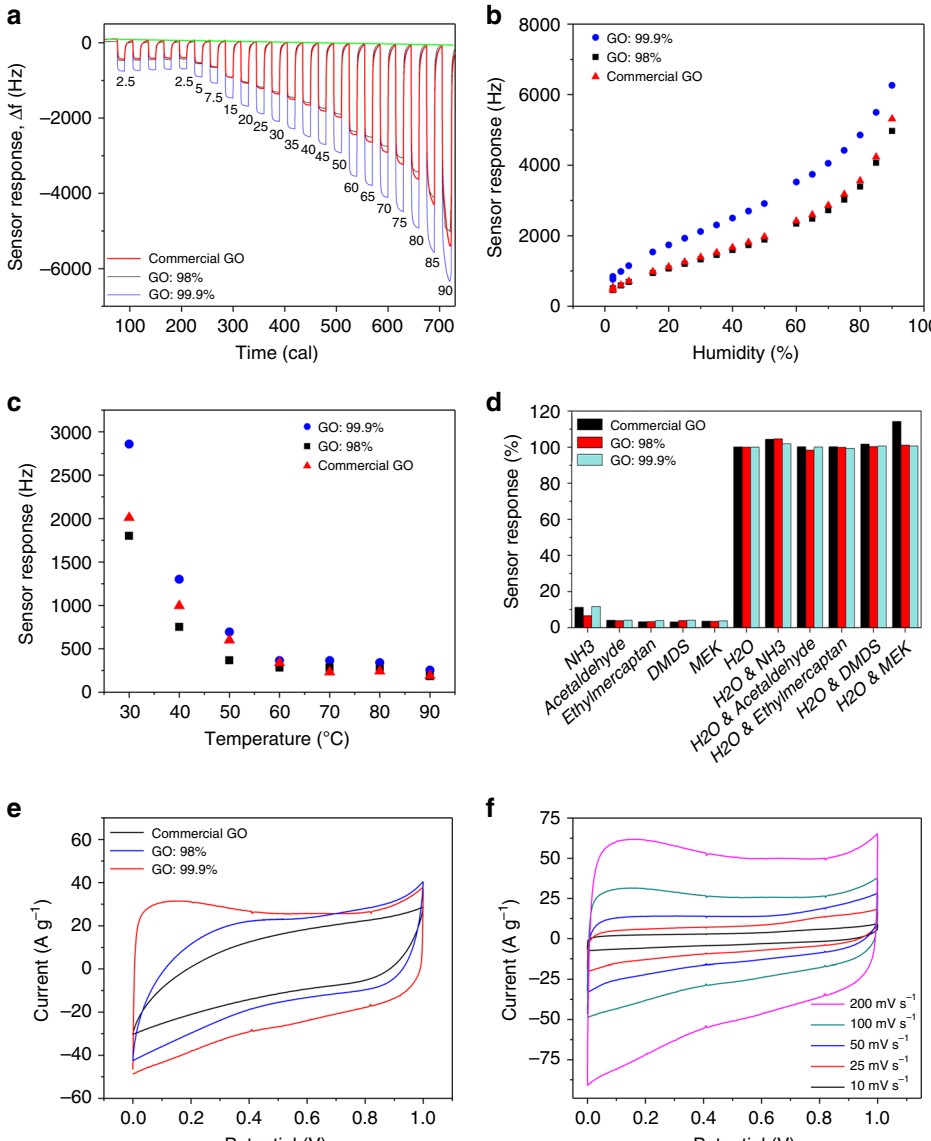

**Fig. 8** The effect of silicon contamination on the device performance. **a** Dynamic response. The green line is the baseline absorption/desorption of a bare Ti-based QCM device. **b** Calibration curve of relative humidity sensors towards humidity concentrations from 2.5 to 90% RH @ 27 °C. **c** The effect of temperature on the sensor response as a function of GO purity (50 % RH). **d** Evaluation of the selectivity of the GO-based sensor (99.9% purity) following exposure to five different interference gases (ammonia, acetaldehyde, ethylmercaptan, dimethyl disulphide and methylethylketone). Following the exposure to the interfering gases, there was no noticeable change in all sensor responses. **e** Double-layer supercapacitor performance of the reduced GO electrodes for the three different materials. The representative cyclic voltammograms (CVs) that were obtained using a two-electrode cell at 100 mV/s and using a 1 M $H_2SO_4$ electrolyte; **f** CV of the rGO electrode made from 99.9% purity graphite as a function of scan rates

and selectivity (Fig. 8d and Supplementary Figure 22), which puts it at the forefront of the best-performing humidity sensors reported in the literature.

GO is an amphiphilic material consisting of both hydrophobic domains (graphenic domains) and hydrophilic parts[27,47]. However, the structure is mainly hydrophilic leading to easy adsorption and adherence of moisture (water) molecules onto the surface[41,48]. Our results show the existence of organo-silicon based contaminants on the surface as a hydrophobic and non-hygroscopic component detrimentally affects the final performance of the sensor leading to lower device performance[49]. Interestingly, the selectivity was not much influenced by the impurity implying that the impurity acts only as a passive barrier. This superb performance can be attributed to the much higher available hydrophilic surface area of the employed GO (99.9%

purity) when the surface contamination is eliminated. This sensors' detection range is shown to be from trace levels to up to >90% RH with unparalleled accuracy and selectivity. Therefore, the sensor developed from high-purity GO can eliminate the need for employing multiple sensors to detect the different humidity levels for any given application. Moreover, as the amount of the material on each sensor is typically around 50 µg, the use of more expensive high-purity precursor does not affect the overall production cost significantly.

To evaluate the effect of silicon-based contamination on the performance of GO after the reduction process, we investigated the double-layer capacitance of reduced GO as a function of contamination. It is a well-established fact that the capacitance behaviour of graphene-based materials is very sensitive to the available surface area[50,51]. Thus the masking effect of the surface

contamination can adversely affect the double-layer charge storage capability. The performance of the three types of reduced GO (rGO) was evaluated. These were compared based on their cyclic voltammetric (CV) responses at 100 mV/s (Fig. 8e). All systems showed a near-rectangular CV curve, except for the commercial rGO. It was evident that the material prepared from high-purity graphite (99.9%) showed a superior capacitive performance and higher electrical conductivity: GO 99.9%: 320 ± 12 s/cm vs GO 98%: 210 ± 13 s/cm and commercial GO: 150 ± 16 s/cm. The near-rectangular CV curves presented in Fig. 8f are representative of excellent double-layer charge storage performance even at high scan rates. Moreover, the maximum capacitance value of 523 F/g recorded at 10 mV/s is close to the theoretical capacitance limit of graphene sheets (550 F/g)[23]. This clearly illustrates that the presence of silicon-based impurities can significantly impair the capacitive performance of graphene sheets.

The findings presented here illustrated how the performance of graphene-based devices is critically dependent on the impurity content, predominantly silicon-based. Furthermore, we showed that the cleaning methods to remove this resilient contamination were not successful, and as such the use of a high-purity precursor in order to obtain high-purity graphene is necessary. Humidity sensors and double layer supercapacitors fabricated with such materials showed significantly improved performance surpassing all existing reported materials and technologies. This emphasises that the silicon-based contamination is a ubiquitous problem in 2D materials produced by exfoliating naturally occurring layered crystals. It also highlights the critical importance of material purity and the need for a quality-control approach to the production and application of 2D-based materials.

## Methods

**GO synthesis**. GO was synthesised with a method described previously[4,27,28,47], using graphite sources of varying purities (shown in parentheses): natural graphite flake (98%), natural graphite powder (99%), natural graphite flake (99.8%), natural graphite flake (99.9%), natural graphite powder (99.9995%), synthetic graphite (99.9995%), and natural graphite powder (99.9999%) from Alfa Aesar. A commercially available GO was also tested as a control. Briefly, graphite powder (1 g) and sulphuric acid (200 mL) were mixed and stirred in a flask for 1 h. Then KMnO$_4$ (10 g) was added to the mixture and stirred for 1 day. The mixture was transferred into an ice bath, and Milli-Q (200 mL) water was added slowly before H$_2$O$_2$ (50 mL) were poured into the mixture. Having stirred for another 30 min, the GO particles were then washed and centrifuged three times with HCl solution (9:1 water/HCl by volume), then centrifuged again and washed with Milli-Q water until the pH of the solution became about 4–5.

**Characterisations**. Specimens for electron microscopy were prepared by deposition of GO suspensions on a holey carbon support film on a copper grid (a droplet of 50 μg/mL). The specimens were stored in glass desiccator to avoid contamination. STEM examination was carried out using an aberration-corrected JEOL ARM200F microscope operating at 80 kV to minimise radiation damage to the specimens. The instrument was fitted with a cold field emission electron source and a JEOL large area (1sr) EDS. This was coupled to a Noran System Seven analytical system. All imaging and analysis was carried in scanning transmission mode (STEM) using a high-resolution imaging probe of approximately 30 pA current and 0.1 nm diameter with a convergence semi-angle of 24.9 mrad. Imaging was carried out in HAADF and BF modes, yielding mass thickness and diffraction contrast information, respectively. The inner and outer acceptance angles for HAADF imaging were 68 and 280 mrad, respectively, and for BF imaging, the acceptance semi-angle was 17 mrad. Scanning images were captured using the Gatan's DigiScan hardware and DigitalMicrograph software.

Scanning electron microscopic (SEM) analysis were carried out by first depositing GO sheets from their dispersions on pre-cleaned and silanised silicon wafer (300 nm SiO$_2$ layer), as described previously[52]. Briefly, silane solution was prepared by mixing 3-aminopropyltriethoxysilane (Aldrich) with water (1:9 v/v) and one drop of hydrochloric acid (Sigma–Aldrich). Precut silicon substrates were silanised by immersing in aqueous silane solution for 30 min and then washed thoroughly with Millipore water. GO sheets were deposited onto silanised silicon substrates by immersing a silicon substrate into the GO dispersion (50 μg/mL) for 5 s and then into a second container containing Millipore water for 30 s and then

air-drying. As-deposited GO sheets were directly examined by SEM (JEOL JSM-7500FA). The lateral size distributions of ~500 isolated GO sheets were determined from several SEM images and analysed using the image analysis software (ImageJ, http://rsb.info.nih.gov/ij/). The lateral size of GO sheets was defined as the diameter of an equal-area circle. Similarly, the size of the graphite particles was measured using optical microscopy.

PL spectra of GO suspensions in water were acquired with a Horiba Jobin Yvon Fluoromax-4 fluorometer with an excitation wavelength of 350 nm. Optical absorption spectra of the same suspensions were obtained on an Agilent Cary60 UV-Vis spectrophotometer. FTIR spectroscopy of GO powders was carried out using a Perkin-Elmer Frontier spectrometer equipped with a Pike GladiATR attenuated total reflectance stage.

XPS analysis was performed using an AXIS Nova spectrometer (Kratos Analytical Inc., Manchester, UK) with a monochromated Al Kα source at a power of 180 W (15 kV × 12 mA), a hemispherical analyser operating in the fixed analyser transmission mode and the standard aperture (analysis area: 0.3 × 0.7 mm$^2$). The total pressure in the main vacuum chamber during analysis was typically between $10^{-9}$ and $10^{-8}$ mbar. Survey spectra were acquired at a pass energy of 160 eV. C 1s high-resolution spectra were recorded at 40 eV pass energy, yielding a typical peak width for polymers of 0.8–1.0 eV. Each specimen was analysed at an emission angle of 0° as measured from the surface normal with an analysis depth of between 5 and 10 nm. Depth profiling experiments were conducted using an Ar Gas Cluster Ion Source (GCIS; Kratos Analytical Inc. Minibeam 6) operated at a cluster size of Ar1000+ with impact energy of 10 keV, equating to partition energy of 10 eV per atom. For the ion beam, a raster size of 1.4 × 1.4 mm$^2$ was employed. WD-XRF spectrometer Model S4 Pioneer, Bruker AXS Gmbh, Karlsruhe, Germany was used to evaluate the average amounts of silicon contamination in the powder form.

**QCM transducer fabrication and characterisation**. The 10 MHz QCM devices were fabricated using optically polished AT-cut quartz substrates (Ø = 7.5 mm) upon the surfaces of which metal electrodes (Ø = 4.5 mm) were e-beam evaporated. The two metal electrodes were each made up of 300 nm of Ti and their sensitivities were calculated to be 4.39 ng/cm$^2$/Hz. Prior to humidity sorption experiments, the mass of GO material deposited on the QCM devices was determined using an Agilent E5100A network. The centre frequency changes of the QCMs were monitored throughout the humidity sorption tests using a Research Quartz Crystal Microbalance (RQCM, Maxtek), which has a frequency resolution of ±0.03 Hz. The QCM response magnitudes were normalised using the mass deposition data in order to obtain a better understanding of the affinity (based on mass) of each material towards humidity.

**Material transfer onto the QCM**. In order to make a RH sensor for testing GO from different sources, a 10 MHz QCM transducer with sensitivity of 4.39 ng/cm$^2$/Hz was employed. Three almost identical thin films of GO were deposited on the QCM devices for testing. Commercially available GO was deposited on the first QCM while the other two consisted of synthesised GO in-house from graphite with purities of 98% and 99.9%. GO dispersions (1 mg/mL) from these sources were drop cast (75 μL) onto the QCM device and were dried at room temperature. The change in the QCM frequency (before and after material deposition) confirmed that similar masses of each GO material were deposited on their respective QCM devices. These deposited mass values were used to normalise the humidity uptake of the materials.

**Humidity uptake measurements**. The humidity uptake measurements were performed in a custom-built environmental chamber, which housed the QCM devices while maintaining a constant operating temperature of 27 °C. The total gas flow rate was kept constant at 200 mL/min throughout the experiments using a multi-channel gas delivery system, employing mass flow controllers (MKS instruments, Inc. USA). The humidity levels were generated using an RH generator (V-Gen from InstruQuest). This humidity level constantly produced by the generator was equivalent to 100% RH at 27 °C or 25.6 g/m$^3$ of water vapour in air. This level of humidity was diluted (by mass flow controllers) to obtain the different concentrations of water vapour required. The humidity sensing experiments throughout the study were such that the sensors were exposed to humidity for 15 min prior to being allowed to recover for a further 15 min under a dry nitrogen atmosphere. The humidity exposure and recovery events combined are referred to as a "pulse" from here on. The signal was determined from the highest humidity exposure (response magnitude) while the noise was determined from a blank profile used in the LOD calculations[53,54].

The selectivity and repeatability experiments were performed with a 50% RH level at 27 °C. Selectivity tests involved exposure to ammonia, acetaldehyde, ethylmercaptan, dimethyl disulphide and methylethylketone with/without the presence of humidity. The contaminant gases and their concentrations were chosen due to their relevance in industrial environments. Repeatability experiments involved the exposure to 50% RH at 27 °C over 10 pulses in a continuous manner. For sensor performance comparison, the response time ($t_{90}$), detection limit, sensitivity and selectivity parameters were used in this study[46,54].

**Double-layer capacitor performance**. The active layer on the electrodes was fabricated through deposition of 5 µL of GO solution (1 mg/mL) on Pt electrodes. The chemical reduction was performed by immersing the electrodes in a solution containing 5 wt% ascorbic acid at 80 °C for 4 h. The double-layer charge storage was investigated using CV experiments using a two-electrode set-up and $H_2SO_4$ (1 M) electrolyte. Please note, in order to highlight the effect of silica contamination (as a barrier) and in order to minimise the effect of restacking of GO sheets on the performance, thin films (<1 µm thickness) of GO were deposited on the electrode. However, for large-scale device fabrication a much higher mass of active material is recommended. This is discussed in the literature[55,56].

## Data availability

The data that support the findings of this study are available from the corresponding authors upon reasonable request.

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

## Acknowledgements

The authors thank the Australian National Fabrication Facility and funding from the Australian Research Council Centre of Excellence Scheme (Project CE 140100012). D.E., R.J., E.D.G. and A.K. acknowledge Vice Chancellor's Fellowship scheme at RMIT University. This research used the JEOL JEM-ARM200F funded by the Australian Research Council (ARC) – Linkage, Infrastructure, Equipment and Facilities (LIEF) grant (LE120100104) located at the UOW Electron Microscopy Centre. S.H.A. acknowledges the financial support from Pasargad Institute for Advanced Innovative Solutions (PIAIS) under Supporting Grant scheme (Project SG1-RMS1705-01) and Equipment and Infrastructure Grant scheme (EI1-MC1709-01). The authors acknowledge Anton Paar GmbH. and Varesh Chimie Bahar Co. for particle size analysis measurements. E.D.G. thanks the ARC for financial support through a DECRA (DE170100164). R.J. thanks the ARC for financial support through a DECRA (DE180100215). The authors acknowledge the facilities and the scientific and technical assistance of the Australian Microscopy & Microanalysis Research Facility at the RMIT Microscopy & Microanalysis Facility, at RMIT University.

## Author contributions

R.J. conceptualised and directed the research project and wrote the manuscript with D.E., S.H.A. and D.R.G.M. R.J. and D.E. were responsible for GO synthesis and XRD, SEM and TEM sample preparation. D.R.G.M carried out all the STEM imaging and EDS analysis with R.J. E.D.G. was responsible for PL, FTIR and UV-Vis analysis. T.R.G. performed the XPS study. Y.M.S. and A.H.K. performed and analysed the sensing data and helped with the manuscript writing. D.E. and R.J. performed and analysed electrochemical device characterisation. A.W. assisted with sample washing process. Y.C., C.W., H.A., D.L.O., D. A.M., S.K.B. and G.G.W. provided support, guidance and proof-read the manuscript; all authors discussed the results and commented on the manuscript.

## Additional information

**Competing interests:** The authors declare no competing interests.

