## [Peer Review File · Nature Communications]

Reviewers' comments:

Reviewer #1 (Remarks to the Author):

The manuscript shows the presence of impurities in graphene, especially silicon and its influence on graphene properties. The presence of impurities within graphene and its impact on its properties is well known (e.g. Tour, Carbon, 2018, 132, 623). It is important to note that the other factors like defect density, composition of oxygen functionalities and, particles size and several other aspects play crucial roles in the chemistry of graphene and its physical properties like capacitance and reaction with surrounding environment. From this point of view the manuscript seems to be not suitable for nature communication.

Reviewer #2 (Remarks to the Author):

This research is very useful and interesting. The author found that the high purity graphene is better for fabricating high performance electronic devices, such as higher performance supercapacitor and humidity sensor.

1. The limit of detection is obtained by testing or calculating?
2. The calculation method of signal-to-noise ratio (in line 256) should be illustrated.
3. I think how to eliminate impurities in graphene will be more meaningful and valuable. The author said that it is almost impossible to eliminate impurities, but how are the different purity graphene obtained? The achieved different purity graphene whether give us some valuable information for eliminating impurities in graphene?
4. The influence of temperature on humidity sensor response should be considered under different purity graphene.

Reviewer #3 (Remarks to the Author):

The authors present a route to reduce a ubiquitous contaminant (silicon atoms) for enhancing the graphene derivatives-based device performance. The authors have focused on the use of high-quality graphite which is a precursor to make graphene derivatives in the experiments to eliminate the contaminant effect in supercapacitor and humidity sensor applications. The silicon contaminant has not been observed in graphene flakes from high quality graphene and the performance of both devices was significantly enhanced when the higher quality graphite used as a precursor. The manuscript is well organized and contains the reasonable discussions for the performed study. The experimental results are quite reasonable for supporting the authors' insist. In these reasons, the manuscript could be published in Nature Communications.

The detailed comments are as follows:

- What is the origin of silicon-based impurities? If the authors may know it, the authors can mention it in the introduction part for the wider readership. Earth abundance of Si or sp³ bonding nature of Si can be discussed.
- Can the authors show the higher atomic resolution of TEM?

x00A0;images for identifying the binding structure between Si and C in the samples?

- In Fig. 6c, the authors can compare the XPS Si 2p

spectra of the Si contaminant with that from a bare Si O₂ substrate to clarify the difference in Si bonding.

- The sheet resistance and optical transmittance are important parameters for graphene derivatives commercialization. The authors can show the sheet resistance changes in the reduced samples from low and high purity graphites.

- English correction is needed. Several typos are found.

Reviewer #1 (Remarks to the Author):

The manuscript shows the presence of impurities in graphene, especially silicon and its influence on graphene properties. The presence of impurities within graphene and its impact on its properties is well known (e.g. Tour, Carbon, 2018, 132, 623). It is important to note that the other factors like defect density, composition of oxygen functionalities and, particles size and several other aspects play crucial roles in the chemistry of graphene and its physical properties like capacitance and reaction with surrounding environment. From this point of view the manuscript seems to be not suitable for nature communication.

With due respect, it is only the presence of metallic impurities on graphene that is well known and is already demonstrated in the literature, including the paper suggested by the reviewer. However, contrary to what the respectful reviewer has suggested, the presence of significant amount of silicon-based impurities, as a previously unknown contaminant in graphene and their significant impact on device fabrication, has not been suggested or investigated prior to this work. Moreover, in order to eliminate the effect of the factors mentioned by the reviewer, samples with almost identical physical and chemical properties were selected. This is supported by XPS, Raman, FT-IR, UV-Vis, SEM and sheet size distribution analysis.

Reviewer #2 (Remarks to the Author):

This research is very useful and interesting. The author found that the high purity graphene is better for fabricating high performance electronic devices, such as higher performance supercapacitor and humidity sensor.

The authors wish to thank the reviewer for recognising the importance and interesting findings presented in this manuscript.

1. The limit of detection is obtained by testing or calculating?

The limit of detection was calculated based on the standard deviation of the sensor noise. The following explanation is now added to supporting information and is highlighted in yellow.

The limit of detection was calculated based on the standard deviation of the sensor noise. This involved calculating the standard deviation of the sensor noise over a 15-minute period (i.e. the exposure time) of three different blank profiles. We then converted this back to humidity concentration using the calibration curve to convert the Δf signal to humidity content¹⁷. This was performed by considering the fact that the sensors exhibited low drift¹⁷. This allows us fit a linear curve for the first lowest 5 humidity concentrations tested.

2. The calculation method of signal-to-noise ratio (in line 256) should be illustrated.

The following explanation is now added to the experimental section and is highlighted in yellow. The signal was determined from the highest humidity exposure (response magnitude) while the noise was determined from a blank profile used in the limit of detection calculations^{50,51}.

3. I think how to eliminate impurities in graphene will be more meaningful and valuable. The author

said that it is almost impossible to eliminate impurities, but how are the different purity graphene obtained? The achieved different purity graphene whether give us some valuable information for eliminating impurities in graphene?

We believe the reviewer is making the point about high-purity graphite in the first place and whether the approach to make high purity graphite is also applicable to graphene production methods. It should be noted although there are many methods to purify graphite such as hydrometallurgical purification methods (acid washing, floatation method) and pyrometallurgical methods (chlorination roasting method and the use of extremely high temperatures, above 2700°C), these methods are not applicable after the exfoliation in liquid media or devices fabrication. In the case of hydrometallurgical purification methods, usually a purification level of up to 98% can be achieved. This method, although very efficient in the removal of metallic impurities, cannot be applied to remove Si-based impurities. On the other hand, heating the graphene at high temperature (more than 2700°C) is not a viable option in our case, as this will limit the versatility of end-product device fabrication. These limitations are much more significant in the case of delicate dispersions of graphene and graphene oxide, as was discussed in the manuscript. As such, we recommend using high-purity graphite (commercially available) in the first place.

This explanation is now given in the main text and is highlighted in yellow.

4. The influence of temperature on humidity sensor response should be considered under different purity graphene.

We have now added figure 7C and S16 to illustrate the temperature dependency of our as-fabricated humidity sensors.

Reviewer #3 (Remarks to the Author):

The authors present a route to reduce a ubiquitous contaminant (silicon atoms) for enhancing the graphene derivatives-based device performance. The authors have focused on the use of high-quality graphite which is a precursor to make graphene derivatives in the experiments to eliminate the contaminant effect in supercapacitor and humidity sensor applications. The silicon contaminant has not been observed in graphene flakes from high quality graphene and the performance of both devices was significantly enhanced when the higher quality graphite used as a precursor. The manuscript is well organized and contains the reasonable discussions for the performed study. The experimental results are quite reasonable for supporting the authors' insist. In these reasons, the manuscript could be published in Nature Communications.

The authors wish to thank the reviewer for recognising the importance and interesting findings presented in this manuscript.

The detailed comments are as followings:

1- What is the origin of silicon-based impurities? If the authors may know it, the authors can mention it in the introduction part for the wider readership. Earth abundance of Si or sp^3 bonding nature of Si can be discussed.

The origin of the silicon impurity and purification techniques are discussed in the main text and can be found below:

Natural graphite is mined and then purified using floatation. The purification in this process is based on differences between the surface chemistry of soil rock and graphite mineral²⁹. However, the floatation process is not able to remove high abundance mineral impurities such as silicon, which are “intercalated” between groups or stacks of adjacent graphene layers. The intercalated impurities are commonly removed by using chemical or thermal treatments³⁰. Graphite particles in the purity range of 80–98 % are typically refined using only floatation. For purities above 98%, additional refinement steps are carried out following floatation³¹. This provides two options to eliminate the contamination: a) purification of the exfoliated materials and b) employing purer graphite precursors.

It should be noted although there are many methods to purify graphite such as hydrometallurgical purification methods (acid washing, floatation method) and pyrometallurgical methods (chlorination roasting method and the use of extremely high temperatures, above 2700°C), these methods are not applicable after the exfoliation in liquid media or devices fabrication. In the case of hydrometallurgical purification methods, usually a purification level of up to 98% can be achieved. This method, although very efficient in the removal of metallic impurities, cannot be applied to remove Si-based impurities. On the other hand, heating the graphene at high temperature (more than 2700°C) is not a viable option in our case, as this will limit the versatility of end-product device fabrication. These limitations are much more significant in the case of delicate dispersions of graphene and graphene oxide, as was discussed in the manuscript. As such, we recommend using high-purity graphite (commercially available) in the first place.

2- Can the authors show the higher atomic resolution of TEM images for identifying the binding structure between Si and C in the samples?

Atomic structural imaging in the STEM works well for bulk crystalline materials. However, for this class of material there are a number of insurmountable challenges. The silicon-containing impurity is amorphous. Clusters of the silicon-containing species are diffuse and there is clearly no defined structural relationship with the underlying graphene. This is in contrast to metal atom clusters (eg Fe). These often develop platelets with a loose, albeit emergent, ordering. The biggest problem is exemplified in Figure S.7, which shows how the specimen is damaged by the electron beam. Aside from holes opening up in the graphene, the adatoms/clusters become highly mobile under the intense electron flux (we use the lowest practical probe current possible 30pA), and this imposes an upper limit on the useful magnification/imaging time at which such features can be imaged/analysed without modifying them. This was despite using the lowest operating voltage for our STEM (80kV) to minimise knock-on damage. Microscope manufacturers are beginning to

develop ultra-low voltage microscopes (30kV) for this type of work, but these are still in the development stage and are not yet available.

Electron energy loss spectroscopy (EELS) in a STEM can provide spectral data with information similar to XPS. However, aside from the issues mentioned above, and the ever present danger of scissioning molecular bonds and changing the valence state due to electron beam-induced reduction, an EELS spectrum with good quality edge information (suitable for fingerprinting against known candidate Si species), would require probably 100x more atoms than are typically present in the features that we are seeing. Such large clusters would of course have a depth of several nm and would therefore include species in contact with carbon, but also, confined within the bulk of the cluster and therefore surrounded by like molecules – producing a hybrid spectrum. Therefore, it is not possible to extract bonding information about any possible Si-C interactions using STEM-based methods on currently available instruments.

3- In Fig. 6c, the authors can compare the XPS Si 2p spectra of the Si contaminant with that from a bare SiO₂ substrate to clarify the difference in Si bonding.

The XPS spectra is now added as Figure S8 to clarify the difference.

4- The sheet resistance and optical transmittance are important parameters for graphene derivatives for commercialization. The authors can show the sheet resistance changes in the reduced samples from low and high purity graphites.

The electrical conductivity (normalised sheet resistance to the thickness) of free standing films made from reduced GO as function of silicon impurity is now added to the main text.

The numbers are as follow:

GO 99.9%: $320 \pm 12 \text{ S.cm}^{-1}$, GO 98%: $210 \pm 13 \text{ S.cm}^{-1}$ and commercial GO: $150 \pm 16 \text{ S.cm}^{-1}$

5- English correction is needed. Several typos are found.

We thank the reviewer for noting the typos. We have now more carefully revised the English of the paper and addressed the typos.

Please feel free to contact me as the corresponding author to discuss any further enquiry.

Reviewers' comments:

Reviewer #1 (Remarks to the Author):

The authors added some few results in the manuscript. I missing the detail analysis of impurities concentration. This have to be provided by ICP / OES or MS with sample digestion or use of NAA which is highly sensitive. This have to be provided for each sample. Many of graphite are in industrial scale washed with HF and silicon oxide or silicate minerals doesn't exhibit typically significant impurities (with exception of metallurgical grade graphite where it doesn't have sense to remove). Some purification has to be demonstrated to see differences in performance like HF digestion which will not significantly affect graphene oxide chemistry. I don't agree that the impurities are "intercalated" in graphene sheets. Intercalation chemistry of graphite as well as its geology and geochemistry and silicon is present as silicon dioxide and silicate minerals as a small particles in graphite matrix, this is definitely not intercalate. X-ray diffraction have to be shown for all materials. Also SAED can be performed on individual impurities accumulation which are visible on TEM images. Why are not provided these data together with EDS distribution maps?

The authors should provide the differences in resistivity of materials.

Important factor is also particle size, this dominantly influencing many properties of GO and graphene. Typically, different graphite has significant differences in particle size distribution. The authors don't provide any particle / sheet size distribution measurement. Some statistic should be provided based on AFM or TEM as well as bulk particle size measurement e.g. by laser scattering method.

The optical properties (mentioned by Referee 3) are not provided.

The STEM in combination with EELS is used at 80 keV for characterization of graphene oxide based materials and I don't see any problem to provide such measurement since it is broadly use to provide detail information about graphene materials.

Reviewer #2 (Remarks to the Author):

I really appreciate the effort done by the authors in order to address my comments and concerns. The paper has been sensibly improved, but one problem still need to be improved. Why the higher purity of graphene can improve the device's performance? In humidity sensor, The author think that the high performance of humidity sensor can be attributed to the much higher available hydrophilic surface area when the surface contamination is eliminated. I think the further and detail explanation is needed for improving the quality of manuscript, especially publication in Nature Communications.

Reviewer #3 (Remarks to the Author):

The manuscript is well-revised and contains useful information for wider readership. I would like to recommend this work to the editor for a publication.

Reviewers' comments:

Reviewer #1 (Remarks to the Author):

The authors added some few results in the manuscript. I missing the detail analysis of impurities concentration. This have to be provided by ICP / OES or MS with sample digestion or use of NAA which is highly sensitive. This have to be provided for each sample.

With due respect, as we have already mentioned in the main text (discussion part line:5-7), “analytical methods, such as ICP-MS, are also not effective as silicon-based compounds (i.e silicon oxides) are unreactive in all acids except HF and all three Si isotopes are subject to N- and O-based interferences.” Furthermore, as mentioned by Sanchez et. al.¹ the ICP-OES signal of silicon depends strongly on the silicon compound and the sample introduction system in use. Thus, for a cross-flow pneumatic nebulizer coupled to a Rytan double pass spray chamber, the variation of the signal can be by a factor of up to 17, depending on the silicon compound. Also, in their study, they showed that matrix effects on silicon cannot be corrected by internal standardization or standards additions. This is shared between both ICP/OES and ICP/MS.

Moreover, ICP/OES or ICP/MS cannot give us any data about the chemical environment of the elements present in the sample. As such, as this is the chemical environment of the Si impurity and its change along the depth which is of utmost interest here, we performed XPS depth profiling, as this data, simply, cannot be derived from any other measurement.

Many of graphite are in industrial scale washed with HF and silicon oxide or silicate minerals doesn't exhibit typically significant impurities (with exception of metallurgical grade graphite where it doesn't have sense to remove). Some purification has to be demonstrated to see differences in performance like HF digestion which will not significantly affect graphene oxide chemistry.

In graphite purification at industrial scale, the final purification step before heat-treatment is using combined sulphuric and hydrofluoric (“HF”) acids to increase the graphite content to more than 99%. However, as we have already mentioned in the revised version of the manuscript, none of these methods are applicable after the exfoliation of the material in liquid media or device fabrication. As is the case with HF or any acids or bases in this regard, their addition to graphene oxide dispersion results in an inevitable agglomeration of graphene oxide sheets which practically hinders the device fabrication process. Therefore, using HF for purification purposes is not recommended. We have now added Figure S3 to supporting information to illustrate the agglomeration process.

Figure S3. Photograph of GO dispersions after extensively washing with Fluoride and NaOH. Purification with such a strong basic solution resulted in an irreversible agglomeration and restacking of GO sheets. Generally, increasing the ionic strength or decreasing the pH of GO suspensions results in loss of the surface charge and restacking of GO particles then occurs.

I don't agree that the impurities are "intercalated" in graphene sheets. Intercalation chemistry of graphite as well as its geology and geochemistry and silicon is present as silicon dioxide and silicate minerals as a small particles in graphite matrix, this is definitely not intercalate. X-ray diffraction have to be shown for all materials.

In the case of exfoliated materials, as we have already mentioned in the main text: “purification with strong basic or acidic solutions results in an irreversible agglomeration and restacking of sheets (Fig. S3). Consequently, the impurities are confined between the layers and remain following the purification process.”

In the case of graphite, although is not in the scope of this manuscript, our STEM analysis supports possibility of intercalation of the impurities between the graphene sheets. As can be seen from the figure appendix 1, both TEM and HAADF images show contaminations (one area is highlighted by the red circle) that are not existent on the surface; image taken by secondary electrons shows the surface only. These impurities are intercalated between the graphene layers or some on the other side.

Figure appendix 1. STEM microscopy analysis of a typical low purity graphite using different imaging modes on a same spot. A) Bright field (BF), B) HAADF image and C) Secondary electron image.

X-ray diffraction of high purity and low purity graphene oxides that are used for device fabrication are now shown as Figure S.17. It should be noted that in the case of sample prepared from lower impurity graphite, the position of the peak slightly shifts towards lower 2θ (higher d -spacing) which can suggest possible trapping of impurities in-between the sheets.

Figure S17. X-ray diffraction patterns of GO films synthesised from low purity (98%) and high purity (99.9%) graphite. Please note, the position of the peak at 2θ of $\sim 11.5^\circ$ shifted very slightly toward the lower 2θ (higher d -spacing) as silica-impurity confined between the sheets.

Also SAED can be performed on individual impurities accumulation which are visible on TEM images. Why are not provided these data together with EDS distribution maps?

Selected area diffraction is a TEM-based technique. It relies on the use of an area-defining aperture to select a region from which to form a diffraction pattern. Due to the inherent uncertainty in the selection of a region for diffraction, the smallest usable diameter of a selected area aperture is around 200nm. Given that the clusters we are describing are atomically dispersed and very small in diameter (1-2 nm), and given that they represent a small fraction of the film – being monolayers in thickness), such features could not be expected to produce any measureable scattering. The silica-rich regions are amorphous, and as such any scattered intensity would be diffuse further exacerbating the problem. At this scale, a more useful approach is the atomic resolution HAADF imaging (STEM-based) method we have used. This allows the atomic locations/single atoms to be directly visualised, and crystalline

materials to be identified by virtue of their periodic structure. Note, an aperiodic structure in an image does not necessarily indicate an amorphous material, since it may simply not be oriented at a strong Bragg condition.

Given that the silica-rich regions we have observed on many dozens of different specimens are all aperiodic it is safe to assume that this phase is amorphous (as one might expect for silica of this nature). One advantage that the HAADF approach is that it does permit sub-areas of an image to be processed using Fourier methods to create a diffractogram. This highlights the periodic structures in that region, providing information which can be analogous to that in a selected area diffraction pattern, albeit from regions as small as several nm in diameter. Clearly, it is possible to process the images we have included in the paper to extract such diffractograms from sub-areas of interest. However, the usual motivation for so doing is to highlight periodic information, often the orientation relationship between two phases. In the case of the silica phase this would not offer any advantage, since the image sub-areas containing silica-rich regions show no periodic information, and therefore, neither would the diffractograms. For the sake of clarity, we have provided such a diffractogram as the appendix 2.

Figure appendix 2. The diffractogram of the selected contaminated area in (A) is presented in B.

Aberration-corrected STEM is possible by virtue of forming an electron probe with a diameter of around 0.1nm. In this work, we were constrained to use an 80kV accelerating voltage, rather than the more typical 200kV beam energy (the latter providing higher spatial resolution). The low voltage helped reduce the knock-on beam damage of the material. Obtaining atomic resolution images at 80kV requires the use of the smallest imaging probe on the instrument. This very small size is achieved at the expense of electron current in the probe, which proportionately reduces the x-ray yield. One must also remember that typical STEM specimens may be 5 to 100nm in thickness, and so the excited volume for x-ray generation is (relatively) large. Here, we are dealing with a material which has sub-nm thickness, resulting in a vanishingly small x-ray yield. For analytical (EDS) work, probes with an order of magnitude more current are typically used – although they are much larger (ca 0.2nm), and typically do not provide atomic resolution. So, the low voltage work necessary to avoid damaging this class of 2D materials necessitates the choice of a probe which enables either atomic resolution or 2D x-ray mapping – but not both (at 200kV it is possible to have both). For this reason, we were constrained to carry out localised point analysis with a low current probe by scanning impurity sub-areas and integrating the EDS spectra over periods of up to 240 s (for a single point). Even if we were prepared to sacrifice resolution and use a larger current probe, another problem is that intense beams mobilise surface atoms and probed clusters simply disintegrate as the atoms/molecules diffuse away from the beam (please see Figure S6). In summary, 2D x-ray mapping of this class of material at 80kV at this scale is not possible.

This explanation is now added as a sub-section in Supporting Information.

The authors should provide the differences in resistivity of materials.

The sheet resistance data normalized to the thickness of the measured films (electrical conductivity) was added to the main text over the previous revision. Please refer to page 12, line 13 of the current manuscript.

Important factor is also particle size, this dominantly influencing many properties of GO and graphene. Typically, different graphite has significant differences in particle size distribution. The authors don't provide any particle / sheet size distribution measurement. Some statistic should be provided based on AFM or TEM as well as bulk particle size measurement e.g. by laser scattering method.

With due respect, we would like to point out that in order to eliminate the effect of the factors mentioned by the reviewer, samples with almost identical physical and chemical properties were selected for device fabrication. This aspect was clearly discussed in the manuscript alongside key comparison between samples. The relevant discussion is in the main text as follow:

“It should be noted that the size of final GO sheets in both samples (contaminated and pure GO) were almost identical (Fig. S13) eliminating the association of the observed phenomena to any size effects. Other physical properties measured by UV-Vis, Raman, FTIR spectroscopy and XRD spectra were almost identical among these GO samples (Fig. S14-17). This, together with the marked similarity in the C 1s spectra (see XPS discussion above) and the aforementioned equal size distribution, confirms that the chemical and physical properties between GO samples are indistinguishable, except for the presence of Si. As we will show in the following section, these silicon-based impurities play a pivotal role in affecting the performance of graphene-based devices.”

For the sake of more clarity: in order to obtain the size distribution, we have obtained SEM micrographs and have measured the area of more than 400 isolated sheets (for each sample). Therefore, we have been able to carry out the complete size characterization of the as-prepared materials. It should also be noted that the only practical technique that could be used to directly observe particle size distribution, in our case, was scanning electron microscopy.

Figure S13. SEM micrographs along with the lateral size distribution comparing GO sheets synthesised from (A-D) low purity (98%) and (E-H) high purity (99.9%) graphite. The lateral size distributions of isolated GO sheets were determined from the SEM images and analyzed using image analysis software (imageJ, <http://rsb.info.nih.gov/ij/>). The lateral size of the GO sheets was defined as the diameter of an equal-area circle. Please note, these two particular graphites contain very similar average grain sizes, which resulted in similar sheet size distribution. Having similar sheet sizes was the motivation to select them for further comparisons and device fabrication. T-test confirmed there was no difference between the mean lateral sizes of both GO samples.

All other techniques such as DLS are sensitive to shape.

As for performing Dynamic light scattering (DLS) studies which is a known method to be used to determine the size distribution profile of small particles in suspensions there are some points that should be considered:

1- First and foremost DLS is limited to a maximum size of $8\mu\text{m}$ where two phases have

similar density and usually less than 600-700 nm for carbon based materials where the

theoretical density is calculated to be 2.2 g cm^{-3} . Therefore, DLS is not suitable for use here.

2- Since DLS essentially measures fluctuations in scattered light intensity due to diffusing particles, it can only calculate the hydrodynamic radius of a spherical particle or at least a 3D particle through the Stokes–Einstein equation and not the real dimension.

3- The hydrodynamic diameter of a nonspherical particle is the diameter of a sphere that has the same translational diffusion speed as the particle. If the

shape of a particle changes in a way that affects the diffusion speed, then the hydrodynamic size will change. For example, small changes in the length of a rod-shaped particle will directly affect the size, whereas changes in the rod's diameter, which will hardly affect the diffusion speed, will be difficult to detect.

4- The DLS calculation is based on Rayleigh scattering and Mie Theory. It implies that if the particles are small compared to the wavelength of the laser used (typically less than $d = \lambda/10$ or around 60nm for a He-Ne laser), then the scattering from a particle illuminated by a vertically polarised laser will be essentially isotropic, i.e. equal in all directions. The Rayleigh approximation tells us that $I \propto d^6$ where I = intensity of light scattered, d = particle diameter.

The d^6 term tells us that a 1000nm particle will scatter 10^6 or one million times as much light as a 100 nm particle. Hence there is a danger that the light from the larger particles will swamp the scattered light from the smaller ones. This d^6 factor also means it is difficult with DLS to measure, say, a mixture of 10000nm and 100nm particles because the contribution to the total light scattered by the small particles will be extremely small. Therefore, the use of DLS here where we have significant fraction of large size sheets and a distribution of large size and small size sheets, would not be appropriate.

The optical properties (mentioned by Referee 3) are not provided.

Detailed optical properties including UV-Vis, Raman, FTIR and PL are already presented and discussed in the main text (Figure 6 e-f) and supporting information as S12, S14, S15 & S16.

The STEM in combination with EELS is used at 80 keV for characterization of graphene oxide based materials and I don't see any problem to provide such measurement since it is broadly used to provide detail information about graphene materials.

The reviewer suggests using EELS, but does not specify which particular question should be addressed by doing so. There are several motivations for doing EELS, which include thickness measurement, compositional analysis

and structural/electronic characterization. Since our focus was on understanding the composition of the impurity regions we selected EDS analysis as the tool of choice. Not only is it simpler and less prone to artefacts than EELS, but it provides information on the entire periodic table from B to U, whereas EELS is best suited to light elements due to their higher scattering cross-sections.

This explanation is now added to Supporting Information.

EELS has been used to provide structural and electronic information on graphene-based materials. However, such studies are beyond the scope of the current work where the focus is not on the graphene oxide *per se*, but rather on the nature of the impurities in graphene oxide, their composition and distribution and their influence on the performance of graphene-oxide-based devices.

Reviewer #2 (Remarks to the Author):

I really appreciate the effort done by the authors in order to address my comments and concerns. The paper has been sensibly improved, but one problem still need to be improved. Why the higher purity of graphene can improve the device's performance? In humidity sensor, The author think that the high performance of humidity sensor can be attributed to the much higher available hydrophilic surface area when the surface contamination is eliminated. I think the further and detail explanation is needed for improving the quality of manuscript, especially publication in Nature Communications.

The authors wish to thank the reviewer for appreciating our efforts to address the comments and suggesting the publication of our paper in Nature communications after addressing the final concern of the reviewer about further explanation on device performance.

We believe the respectful reviewer's concern is about the humidity sensor's performance. It should be noted that the humidity sensors were made out of graphene oxide and not "graphene".

The following explanation is now added to the main text.

Graphene oxide is an amphiphilic material consisting of both hydrophobic domains (graphenic domains) and hydrophilic parts.^{2,3} However, the structure is mainly hydrophilic leading to easy adsorption and adherence of moisture (water) molecules onto the surface.^{4,5} Our results show, the existence of organo-silicon based contaminants on the surface as a hydrophobic and non-hygroscopic component detrimentally affects the final performance of the sensor leading to lower device performance.⁶

Reviewer #3 (Remarks to the Author):

The manuscript is well-revised and contains useful information for wider readership. I would like to recommend this work to the editor for a publication.

We thank the reviewer for the recommendation of the publication of the manuscript without change.

References:

- 1 Sánchez, R., Todolí, J.-L., Lienemann, C.-P. & Mermet, J.-M. Effect of the silicon chemical form on the emission intensity in inductively coupled plasma atomic emission spectrometry for xylene matrices. *Journal of Analytical Atomic Spectrometry* **24**, 391-401,(2009).
- 2 Aboutalebi, S. H. *et al.* Comparison of GO, GO/MWCNTs composite and MWCNTs as potential electrode materials for supercapacitors. *Energy & Environmental Science* **4**, 1855-1865,(2011).
- 3 Aboutalebi, S. H., Gudarzi, M. M., Zheng, Q. B. & Kim, J. K. Spontaneous Formation of Liquid Crystals in Ultralarge Graphene Oxide Dispersions. *Advanced Functional Materials* **21**, 2978-2988,(2011).
- 4 Gao, W. *et al.* Direct laser writing of micro-supercapacitors on hydrated graphite oxide films. *Nature Nanotechnology* **6**, 496,(2011).
- 5 Liu, R., Gong, T., Zhang, K. & Lee, C. Graphene oxide papers with high water adsorption capacity for air dehumidification. *Scientific Reports* **7**, 9761,(2017).
- 6 Xu, Z.-W. *et al.* Enhancement on the Surface Hydrophobicity and Oleophobicity of an Organosilicon Film by Conformity Deposition and Surface Fluorination Etching. *Materials* **11**, 1089,(2018).

Please feel free to contact me as the corresponding author to discuss any further enquiry.

Reviewers' comments:

Reviewer #1 (Remarks to the Author):

The authors provide reply on several referee questions. Regarding this the laser diffraction measurement is standardly used for average size distribution of particles in the range of 10 nm to over 1mm. The authors probably misunderstand the reviewer report. I still missing any elemental composition information. The authors not provide ICP, but in the material can be many other elements responsible for change of the properties. I believe that sample combustion and digestion with HF will provide relevant data since the silicon will be in form of H_2SiF_6 which is relevant for ICP analysis and can be calibrated. Many analytical methods are suitable for elemental analysis, like X-ray fluorescence, which can give proper information about elemental composition from fluorine to more heavy elements. Many other methods can be found for silicon quantification. Why is not provided survey XPS spectra and relevant composition calculation from them? The changes in X-ray diffraction are so small that doesn't say anything. Such small deviation can originate from instrument error (differences in vertical placement of sample in few microns or just change of humidity since water simply intercalate in graphene oxide. I don't see the manuscript suitable for publication in Nat Commun.

Reviewer #2 (Remarks to the Author):

This manuscript has been well revised, I would like to recommend the editor for publication.

Answer to the reviewer:

The authors provide reply on several referee questions. Regarding this the laser diffraction measurement is standardly used for average size distribution of particles in the range of 10 nm to over 1mm. The authors probably misunderstand the reviewer report.

The light techniques for doing these measurements are categorized into two groups, namely dynamic light scattering (covering the range from 0.3 nm to 8 um) and static light scattering (from 1 um to almost over 1mm). For both Fraunhofer's and Mie's approximations methods used in Light scattering techniques, particles are assumed to be *spherical*, which is not the case with any 2D materials. Irregular shape 3D particles can therefore be used for this method with some errors, however, this method cannot be applied to any 2D materials. There are also many other false assumptions such as random particle orientation assumption. This assumption, as also discussed by Kerry and etzler (attached reports), is false regarding non-spherical particles, because the particles will orient themselves in the direction of flow. This phenomenon, on its own, proves that the random orientation assumption in our case is false. Moreover, "Dynamic light scattering measures the translational diffusion coefficient of particles, assuming that they are spherical. In a very simplified way, the Stokes Einstein equation is applied to calculate the size of the particles. Because graphene does not have rotational symmetry, two translational diffusion coefficients are measured and also the rotational diffusion coefficient impacts on the measurement results. When looking at the measured data, one can see that in this case DLS is not the appropriate measurement technique for this sample. "This quotation comes from one reputable manufacturer's report on our own sample, Anton Paar (attached report). We asked Anton Paar to perform the tests for us and in conclusion they came up with this statement, Page 4 of the customer report under conclusion. For the sake of clarity and transparency, we have also included this report as a separate .pdf file to this letter. If someone also consults with any other manufacturer's reports (such as Horiba, Malvern and many others), he/she will see the same claim that the light scattering technique for extremely non-spherical particles especially with large aspect ratio over 5 (in our case the aspect ratio is almost over 15,000) is not a suitable method.

We have also attached the guideline from Horiba specifically stating, "The only techniques that can describe particle size using multiple values are microscopy or automated image analysis. An image analysis system could describe the non-spherical particle seen in Figure 1 using the longest and shortest diameters, perimeter, projected area, or again by equivalent spherical diameter. When reporting a particle size distribution, the most common format used even for

image analysis systems is equivalent spherical diameter on the x axis and percent on the y axis.” The guideline from Horiba is also attached as a separate .pdf file to this document.

Therefore, the only practical technique that could be used to directly observe and analyze particle size distribution, in our case, was microscopy techniques.

In the revised manuscript, we have provided size distribution of graphite flakes and resultant GO in Figure S15 and Figure S16, respectively.

Figure S15. Representative micrographs of graphite flakes along with flake size distribution of (A-C) low purity graphite (98%) and (B-F) high purity graphite (99.9%) used in this study to synthesize GO for the device fabrication. Graphite flakes with similar mesh (-20+80) were used to assure synthesis of GO with analogous sheet sizes distribution (Figure S16). The lateral size of each graphite flake was determined from an optical microscopy image and analyzed using image analysis software (ImageJ, <http://rsb.info.nih.gov/ij/>). The lateral size of the graphite flake was defined as the diameter of an equal-area circle. T-test confirmed there was no difference between the mean lateral sizes of both graphite samples.

Figure S16. SEM micrographs along with the lateral size distribution comparing GO sheets synthesized from (A-D) low purity (98%) and (E-H) high purity (99.9%) graphite. The lateral size distributions of isolated GO sheets were determined from the SEM images and analyzed using image analysis software (ImageJ). The lateral size of the GO sheets was defined as the diameter of an equal-area circle. T-test confirmed there was no difference between the mean lateral sizes of both GO samples.

I still missing any elemental composition information.

With due respect, we have already provided complete XPS analysis and elemental composition of the bulk material used for the devices (Figure 6, S11, S12 and S13). Furthermore, XPS survey along with compositional analysis are now added as Figures S9 & S10.

Figure S9. XPS survey of GO from low purity graphite (98%). Please note the observed elements are Si (0.2 atomic %), C (70.6 atomic %) and O (29.2 atomic %). We should point that out we have observed traces of Cl and S in some earlier samples (residue from H₂SO₄ treatment and HCl washing process, respectively). Cl and S were removed effectively by

further washing and purification. However, the Si impurity was not removed over the purification process.

Figure S10. XPS survey of GO from low high graphite (99.9%). Please note the observed elements are C (70.3 atomic %) and O (29.7 atomic %).

The authors not provide ICP, but in the material can be many other elements responsible for change of the properties. I believe that sample combustion and digestion with HF will provide relevant date since the silicon will be in form of H_2SiF_6 which is relevant for ICP analysis and can be calibrated.

Apart from the discussed technical challenges for measuring Si using ICP (for example all three Si isotopes are subject to N- and O-based interferences), digestion with HF is not along with our safety standards implemented in the labs. We should strictly limit the use of HF, as it is regarded as a highly hazardous substance with severe environmental and health impact. Thus, we have obtained both the elemental composition and chemical environment data using XPS, which was much more useful for our study compared to ICP.

Likewise, high angle annular dark field (HAADF) imaging combined with energy dispersive x-ray spectroscopy (EDS) in an aberration-corrected scanning transmission electron microscope (STEM) was employed to directly detect and monitor the slightest change in the material.

Furthermore, burning GO at high temperature may result in more contamination as most of the crucibles and tube furnaces contained silica.

but in the material can be many other elements responsible for change of the properties.

We have carefully produced the final devices using material with almost identical physical and chemical properties (except silicon impurity).

At the single atomic level:

Any molecularly-dispersed contamination was analyzed using atomic resolution capabilities of aberration-corrected STEM single atom imaging in HAADF mode by virtue of its high atomic number sensitivity. The impurities were also quantified by performing EDS in parallel with the HAADF on the same region.

After a very careful sample preparation and over multi-step purifications, the microscopy and EDS spectrum of GO material identified only silicon-based contamination and very few scattered atoms of Ca from the water (background in every sample). As can be seen (Fig.1 f-g & S1), the peaks at ~0.277, 0.525 and 1.739 keV in the EDS spectrum are due to C, O and Si, respectively while, the peak at 0.930 keV is from the Cu (support) grid. Comparing the EDS spectra of two neighbouring regions, one clean (dark) and the other bright (contaminated) confirms silicon to be the contaminant (Fig.1 f-g & S1). The contaminated region (red boxed region in Fig. 1d) showed a noticeable peak at 1.739 keV (Fig. 1 f), while the clean regions (green box in Fig. 1d) showed no such silicon peak (Fig. 1 g).

Figure 1. The extent of silicon-based contamination on the surface of typical graphene oxide derived from low purity graphite (98 % purity). a) Bright field (BF) image of a typical GO sheet. b) HAADF image of (a). c-d) Details BF and HAADF images of the marked region in (a) at higher magnification, respectively. Unlike the BF images in which Si contaminants are largely invisible, the HAADF images highlights them as bright clusters. e) EDS spectrum of the entire region shown as pink box in (a & c). The strong Si peak at 1.739 keV confirms the significant contamination in the GO sample. f-g) A comparison of the EDS spectra of the contaminated area (f) and non-contaminated area (g), which are marked as red and green boxes in (d), respectively.

In the case of the bulk GO/rGO films used for the device fabrication:

In order to eliminate the effect of the factors mentioned by the reviewer, precise measurements were performed on the samples to obtain almost identical physical and chemical properties (except Si contamination). This was supported by XPS, Raman, FT-IR, UV-Vis, SEM and sheet size distribution analysis.

Many analytical methods are suitable for elemental analysis, like X-ray fluorescence, which can give proper information about elemental composition from fluorine to more heavy elements. Many other methods can be used for silicon quantification. Why is not provided survey XPS spectra and relevant composition calculation from them?

We have provided complete XPS analysis and elemental composition at Figure 6, S11, S12 and S13 as well as S9 & 10 that are added to this revision.

We also note, as graphene is essentially surface with no bulk, surface sensitive analysis such as XPS was more favourable in our study than bulk elemental characterisation techniques such as X-ray fluorescence (XRF).

The changes in X-ray diffraction are so small that doesn't say anything. Such small deviation can originate from instrument error (differences in vertical placement of sample in few microns or just change of humidity since water simply intercalate in graphene oxide).

Although, XRD is not an appropriate technique for identifying atomically dispersed impurities, we have reported the results in the supporting information because the reviewer, in the first place, had asked about the XRD data. Even though we didn't draw any conclusions from this data, the instrument was calibrated carefully using NaCl crystals. Here, the measurements were performed using an auto-sampler unit and then without changing the samples or getting them out, the NaCl crystals were placed on top of the samples. Then, we checked for any deviations in the peak position, if there is any.

REVIEWERS' COMMENTS:

Reviewer #1 (Remarks to the Author):

The authors provide suitable explanation for the question regarding particle size and X-ray diffraction. But I don't understand why it is not possible to provide "average" concentration of silicon using some standard method like WD-XRF or ICP-OES. In the material is relatively high concentration of silicon on the sheet surface, but it can be significantly different from average concentration under few layers of sheets. For such concentration is not necessary to use methods like ICP-MS, since concentration should be relatively high according to the authors statements.

Reviewers' comments:

Reviewer #1 (Remarks to the Author):

The authors provide suitable explanation for the question regarding particle size and X-ray diffraction. But I don't understand why it is not possible to provide "average" concentration of silicon using some standard method like WD-XRF or ICP-OES. In the material is relatively high concentration of silicon on the sheet surface, but it can be significantly different from average concentration under few layers of sheets. For such concentration is not necessary to use methods like ICP-MS, since concentration should be relatively high according to the authors statements.

As the reviewer requested, the average concentration of silicon in the bulk sample has been measured using WD-XRF. This result that showed 0.04 ± 0.007 % and 0.25 ± 0.01 % silicon in the pure and non-pure samples, respectively, was in agreement with our original XPS depth profiling analysis.

The flowing description is added to the manuscript:

Results section:

In order to evaluate the average amounts of silicon contamination in the bulk materials, Wavelength Dispersive X-ray Fluorescence (WD-XRF) spectroscopy was used. Analogues to the XPS depth profiling measurement, 0.04 ± 0.007 % and 0.25 ± 0.01 % silicon were found in the pure and non-pure samples, respectively.

Experimental section:

WD-XRF spectrometer Model S4 Pioneer, Bruker AXS GmbH, Karlsruhe, Germany was used to evaluate the average amounts of silicon contamination in the powder form.

Please feel free to contact me as the corresponding author to discuss any further enquiry.